# Infections of the *Xylella fastidiosa* subsp. *pauca* Strain “De Donno” in Alfalfa (*Medicago sativa*) Elicits an Overactive Immune Response

**DOI:** 10.3390/plants8090335

**Published:** 2019-09-07

**Authors:** Raied Abou Kubaa, Annalisa Giampetruzzi, Giuseppe Altamura, Maria Saponari, Pasquale Saldarelli

**Affiliations:** 1CNR, Institute for Sustainable Plant Protection, via Amendola 122/D, 70126 Bari, Italy; 2Department of Soil Science, Plants and Foods, University of Bari, via Amendola 165/A, 70126 Bari, Italy

**Keywords:** *Xylella*, alfalfa, medicago, transcriptome, olive, *pauca*, immune response, necrosis

## Abstract

Diseases caused by *Xylella fastidiosa* are among the most destructive for several agricultural productions. A deadly disease of olive, termed olive quick decline syndrome, is one of the most recent examples of the severe impacts caused by the introduction and spread of this bacterium in new ecosystems with favorable epidemiological conditions. Deciphering the cascade of events leading to the development of severe alterations in the susceptible host plants is a priority of several research programs investigating strategies to mitigate the detrimental impacts of the infections. However, in the case of olives, the long latent period (>1 year) makes this pathosystem not amenable for such studies. We have inoculated alfalfa (*Medicago sativa*) with the olive-infecting strain “De Donno” isolated from a symptomatic olive in Apulia (Italy), and we demonstrated that this highly pathogenic strain causes an overactive reaction that ends up with the necrosis of the inoculated stem, a reaction that differs from the notoriously Alfalfa Dwarf disease, caused by *X. fastidiosa* strains isolated from grapes and almonds. RNASeq analysis showed that major plant immunity pathways are activated, in particular, several calcium transmembrane transporters and enzymes responsible for the production of reactive oxygen species (ROS). Signs of the necrotic reaction are anticipated by the upregulation of genes responsible for plant cell death and the hypersensitive reaction. Overall the whole infection process takes four months in alfalfa, which makes this pathosystem suitable for studies involving either the plant response to the infection or the role of *Xylella* genes in the expression of symptoms.

## 1. Introduction

Among the different classes of plant pathogens, the bacterium *Xylella fastidiosa* accounts for the largest host range [1] and is responsible for some of the most destructive plant diseases (i.e., Pierce disease and olive quick decline syndrome “OQDS”). Even so, detrimental infections are restricted to a limited number of susceptible plant species, suggesting that, in the majority of the hosts, *X. fastidiosa* most likely acts as commensal, becoming highly pathogenic when specific conditions and host interactions occur. Virulence and other biological characteristics (i.e., growth, transmissibility, and host range) are highly variable depending on the genetic traits of the strain and on the plant host response [2]. Mechanisms underlining host–bacterium interactions and particularly those that induce a shift from the asymptomatic stage of the infections to the expression of symptoms are not well understood, even if, in general, *Xylella*-induced diseases recall water stress, a condition determined by the vascular occlusions caused by the pathogen’s biofilm formation and by the host responses (i.e., the production of tyloses and gums) [3]. However, such a simple mechanism has been repeatedly questioned by several observations, in which, for example, grapevines undergoing water stress have different gene expression profiles than plants infected by *X. fastidiosa* [4]. Deletion of the bacterial rpfF genes, while its ability to form biofilm is reduced, still causes more virulent infections and, nevertheless, the discovery of “virulence” factors [5]. Successful infection of the host requires that the bacterium efficiently replicates and colonizes the xylem vessels, whose integrity is altered by several bacterial secreted enzymes degrading polysaccharide (polygalacturonase A), protein (protease A), and lipid (lipase/esterase LesA) constituents of the plant cell wall [6]. As from the host side, an increasing number of studies suggest that an immune response is indeed triggered by *X. fastidiosa*. In grapevines, Rapicavoli et al. [7] reported that, at the initial stage of the infections, lipolysaccharide O-antigen acts as a sort of “shield” masking the bacterial cells from triggering the grapevine immune system. Similarly, several receptor-like kinases (RLKs) known to interact with pathogen-associated molecular patterns (PAMPs) or microbe-associated molecular patterns (MAMPs) are upregulated during the infection processes established by different strains able to colonize citrus [8] and olives [9]. 

Olive quick decline syndrome (OQDS) is one of the most recent examples of the severe outcomes resulting from the introduction of this bacterium in new areas with favorable epidemiological conditions [10]. Aetiological studies on the OQDS showed that isolates of *X. fastidiosa* subspecies *pauca* ST53 are the causative agent of this disease, decimating olive trees in one of the main olive growing area of Southern Italy (the Apulia region). Given the novelty and the lack of knowledge on the olive–*Xylella* interactions, a multitude of research investigations has been stimulated at the EU level, whose preliminary results clearly demonstrate the high susceptibility of different olive cultivars to *Xylella* as well as the long incubation period of the infections on olives, with symptoms appearing no earlier than a year after the artificial or vector-mediated inoculation. Although a shorter latency period was recorded for the same olive strain when inoculated on oleander (*Nerium oleander*) and *Poligala myrtifolia* [11], the need to have a suitable host that can serve as a model pathosystem not only to study the pathogenicity and host defense mechanisms but also to test different therapeutic applications is amongst the most pursued research objectives. To this end, we have made several attempts to infect annual (*Chenopodium* spp., *Erigeron* spp. *Nicotiana tabacum* SR1) and perennial plant species with the olive reference strain “De Donno” of *X. fastidiosa*, subspecies *pauca* ST53 [12]. Among the selected species we successfully needle-inoculated alfalfa (*Medicago sativa*), whose systemic infections resulted in initial necrotic reactions of the inoculated stem in the likely attempt to isolate and restrict the spread of the bacterium in the plants. Previous reports of *Xylella* infections in alfalfa refer to strains of the subspecies *fastidiosa* causing the typical phenomena of dwarfing, and the relevance of the infections relates more to the epidemiological role of the infected alfalfa crops as a reservoir of the inoculum for the insect vectors spreading the bacterium in the surrounding vineyards [13] rather than to the direct damage to the crop. To investigate molecular alterations occurring during the onset of this severe response a transcriptome analysis of these alfalfa-infected tissues was undertaken revealing the insurgence of a strong immune reaction. 

## 2. Results

### 2.1. X. fastidiosa “De Donno” Elicits a Necrotic Response in Alfalfa

For both rounds of experiments, four months after the inoculations, necrosis and dieback was recorded on the inoculated stem of some plants (Figure 1). These necrotic alterations remained restricted to the inoculated shoots. 

Sampling from these inoculated shoots at three months post inoculation (mpi) allowed for the detection of the bacterium in 6 out of 13 plants from the first experiment and in 7 out 14 plants in the second experiment. At 6 mpi, when sampling was extended to the non-inoculated shoots, results showed limited systemic colonization: only one plant tested positive in the first experiment and 7 plants in the second experiment. On these positive plants, sampling of roots was immediately performed, showing the presence of the bacterium only in four plants of the second experiment, whereas a lack of bacterial detection occurred in the single infected plant of the first experiment. In these four systemically infected plants, other than the desiccation of the inoculated shoots, no additional specific symptoms were observed up to one year after the inoculation. 

Symptoms of leaf scorching (Appendix A) were occasionally recorded on the plants used in the first experiment, but no positive correlation was found with the presence of the bacterium in the symptomatic plants. 

### 2.2. Mapping to the M. sativa Transcriptome

Due to the complex tetraploid genome of *Medicago sativa*, available genome information is lacking. Therefore, genetic studies rely only on a de novo transcriptome assembly (MSGI 1.2; [14]) obtained from the subspecies *sativa* (B47) and *falcata* (F56), consisting of 112,626 unique sequences from different plant organs and tissues. While a complete genome of the close relative *Medicago truncatula* (GCF_000219495.3_MedtrA17_4.0) can be used as a model species for studies on legume genomics [15]. 

Three alfalfa plants infected with *Xylella fastidiosa* “De Donno” (Table 1) that showed symptoms of necrosis of the inoculated stems were used for the RNASeq analysis. Tissues consisted of stems and leaves from the plant green sectors on which preliminary qPCR tests confirmed the infections. Similar tissues from three uninfected and mock-inoculated seedlings were used for healthy control comparison. Numbers of obtained left and right sequenced reads ranged between 17.7 to 22.7 million, of which, on average, 75% were mapped to the MSGI 1.2 transcriptome database [14]. When aligned in pair, the alignment rate of concordant reads was more that 61% (Appendix A, *M. sativa*). 

Relationships among plants status (i.e., infected and healthy) and biological replicates were exemplified by principal component analysis (PCA), which clearly separates Xf-infected from healthy plants (Figure 2a). Principal Component 1 and 2 respectively represent 38 and 26% of the total variance, showing a compact clustering of healthy plants and a wider distribution of the three infected plants. 

DESeq 2 statistical analysis identifies 2300 differentially expressed genes (DEGs) distributed among 1550 upregulated and 750 downregulated (Figure 3a, Appendix A, *M. sativa*). Four hundred seventy-eight (478) DEGs have “no description” when searched in the Alfalfa Gene Index and Expression Atlas Database (http://plantgrn.noble.org/AGED/Download.jsp), a contig annotation database—in particular, 305 are among the 1550 upregulated DEGs. In depth BLASTN analysis of the first 20 upregulated transcripts showed that the third (contig_3095 in Appendix A, *M. sativa*) shows similarity to the mRNA of *M. truncatula* coding for the TMV resistance protein N isoform X1, a well characterized disease resistance protein (Figure 3b). It interacts directly or indirectly with avirulence proteins and triggers a defense response that can culminate with a hypersensitive response to restrict the pathogen growth and movement [16]. The plant needs to maintain a correct redox homeostasis is shown by the upregulation of a protein of the Thioredoxin superfamily (contig_3716 in Figure 3b and Appendix A, *M. sativa*), which is particularly expressed during development processes and responses to pathogen infections. Besides protecting from oxidative damages, these proteins act as signaling for the plant immune system [17]. The upregulation of a zinc transporter (contig 13435 in Figure 3b and Appendix A, *M. sativa*) is reminiscent of the alteration of the leaf ionome in tissues of plants infected by *X. fastidiosa* [18], a condition which is highly represented by the presence of many genes involved in calcium level regulation (i.e., contig_98743, a calcium-dependent, lipid-binding, CaLB domain family protein, in Figure 3b and Appendix A, *M. sativa*) and or calcium signaling (contig_7666, calcium-dependent protein kinase 28; contig_10527, calmodulin-domain protein kinase CDPK isoform 2; contig_33703, calmodulin 5). 

Similarly, many genes that lack annotation were among the highest downregulated transcripts. Besides a protein containing a domain of unknown function (DUF21; contig_63219), a ribosome translation elongation factor EFG/EG2 (contig_63219) was strongly downregulated, indicating the need to tightly control protein synthesis. Contig_64504, encoding a concanavalin A-like lectin protein kinase family was also downregulated. This protein belongs to L-type lectin receptor kinases, which take part in plant immunity and are differentially involved in resistance or susceptibility depending on the pathogen [19] (Figure 3b).

Pathway analysis and transcript categorization was further accomplished by MapMan, which allowed for the identification of 619 DEGs of which 417 were mapped in the range +2 < fold change < −2. The majority of upregulated transcripts (129) were not classified followed by pathways containing signaling (43), development (18), RNA (18), transport (23), and protein metabolism (17) transcripts, all having statistical significance at *p* < 0.05 (Appendix A, total mapping *M. sativa*). Similarly, downregulated transcripts having significant altered expression were classified as not assigned (119) or were classified in the miscellaneous (39), secondary metabolism (28), lipid metabolism (18), protein metabolism (10), stress (11), and RNA (14) pathways (Appendix A, total mapping *M. sativa*).

In depth analysis of upregulated transcripts in the signaling pathway identified several contigs (contig_103371, contig_17750, contig_11416, contig_46853, contig_94096, contig_65187, contig_83845, contig_88907, and contig_60309 in Figure 3b and Appendix A, Signaling, *M. sativa*) corresponding to orthologues of receptor kinases mostly represented by homologues of the leaf rust 10 disease-resistance locus receptor-like protein kinase. Besides being involved in rust resistance in wheat, this gene was shown to mediate ABA signaling and confer drought resistance in *A. thaliana* [20]. A further populated category of signaling proteins was involved in transducing Ca++ signals, particularly of the abscisic acid pathway (contig_10527 and contig_6738, Figure 3b and Appendix A, Signaling, *M. sativa*) or are transmembrane ion transporters devoted to maintain Ca++ cytoplasmic homeostasis (contig_68437, contig_45401, contig_94087, and contig_26528, in Figure 3b and Appendix A, Signaling, *M. sativa*). Evidences of the cell death and necrosis phenomenon are shown by the upregulation of a lipid acid hydrolase belonging to the patatin-like gene family (PLP) (contig_7526, contig_58886, contig_65157, and contig_93785 in Figure 3b and Appendix A, Development, *M. sativa*) [21]. These proteins and particularly PLP2 are overexpressed in response to fungal and bacterial pathogens and enhance plant cell death during *Botrytis cinerea* infections in *A. thaliana*. 

Among the strongly (up to −4.35 fold expression) downregulated transcripts, those with phosphatase activities are the majority (Appendix A, Miscellaneous, *M. sativa*). Indeed a possible mechanism involving inorganic phosphate (Pi) in response to *X. fastidiosa* has been postulated in the grapevine [22] and has been suggested in the expression of HLB symptoms [23]. In both pathosystems, lower concentrations of Pi were observed in infected xylem sap (grapevine) or plant tissues (citrus) as well as the upregulation of specific microRNAs (miR399 and miR828) induced by Pi starvation. An alternative hypothesis to explain phosphatase downregulation relies on the plant need to control excess phosphorelay signaling, a two component system in which a sensor kinase transfer a phosphate group to a cognate response regulator [24]. This system, activated by the perception of a pathogen infection (a PAMP or a MAMP) generally by a cell surface receptors, needs to be finely regulated by phosphatases to contain an excess activation, which can be deleterious for the plant cells. In a more general view, phosphatases contribute to maintain immune homeostasis and to finely tune the amplitude of the plant immune response, which mainly relies on the phosphorylation status of different immune components [25].

A member of the cytochrome P450 monooxygenases (CYP) family binning in the secondary metabolism pathway was found strongly downregulated in infected plants (at4g31500 in Appendix A, Secondary Metabolism, *M. sativa*). This class of enzymes plays a major role in plant defense against pathogens and insects because of its involvement in the glucosinolates biosynthesis. The downregulation of a member of this family is reported to occur in PD-susceptible cultivars [26]. 

Overall, enriched pathways after mapping to *M. sativa* transcriptome delineate an intense stimulation of the plant defense system, as can be found by the differential expression of transcripts related to signaling, Ca++ ion transmembrane transport, oxidation/reduction, and secondary metabolisms. 

### 2.3. Mapping to the M. truncatula Genome

Differential gene expression analysis was also carried out on the assembled genome of *M. truncatula*, which is used to facilitate studies on legume species [15]. As expected, a lower number of reads, from 35.8 to 39.1% of the total, were mapped to the *M. truncatula* genome, with respect to those aligned to the *M. sativa* transcriptome (Appendix A, *M. truncatula*). Relationships among libraries and consistency of the replicates was shown by PCA in which Principal Component 1 and 2 respectively explain 54 and 26% of the total variance and clearly distinguish healthy from *Xylella*-infected plants (Figure 2b). Among the 57,585 mRNAs (XM_#) annotated in the MedtrA17_4.0 version of the *M. truncatula* genome (ftp://ftp.ncbi.nlm.nih.gov/genomes/all/GCF/000/219/495/GCF_000219495.3_MedtrA17_4.0/GCF_000219495.3_MedtrA17_4.0_genomic.gff.gz), a total of 1033 mRNAs were found differentially expressed by DESeq2 analysis and were distinct in 748 up- and 285 downregulated whose minimum and maximum log2 fold change were −4.67 and +3.49, respectively. Only DEG >+1 and <−1 are considered for further analysis (Figure 4a and Appendix A, *M. truncatula*). 

DESeq2 analysis identified Monothiol glutaredoxin-S6 (GRX S6) as the upregulated gene (XM_003625826.2) having the highest fold change (Figure 4b and Appendix A, *M. truncatula*). GRX6 contributes to the maintenance of cellular redox homeostasis during pathogen infection because of its role in scavenging ROS that are detrimental if excessively produced. Moreover, maintaining cellular redox homeostasis regulates posttranslational modifications of target proteins involved in organ development and defense responses against biotic and abiotic stresses [27]. 

A basic 7S globulin 2 is the second upregulated protein (XM_013607298.1) (Figure 4b and Appendix A, *M. truncatula*). It is hortologous to a xyloglucan-specific endo-β-1,4-glucanase inhibitor protein (XEGIP) in tomatoes [28] that inhibits fungal xyloglucan-specific endo-β-1,4-glucanase (XEG). We may assume that this specific plant response is likely consequential to the cell-wall degrading activity of *Xylella*-secreted enzymes.

Zinc transporter 2 (ZIP2) (XM_003597339.2) belongs to a family of transmembrane transporter promoting zinc uptake in the cytoplasm [29] (Figure 4b and Appendix A, *M. truncatula*). The role of zinc in *Xylella* infections is not completely clear as it was found increased [30] or decreased [31] in citrus or unaltered in *Nicotiana tabacum* [18]. The upregulated metal-nicotianamine transporter YSL3 (XM_003597339.2) is a regulator of the salicilic acid/jasmonic acid signaling pathways whose knockout mutant makes plants susceptible to *Pseudomonas syringe* pv. tomato (Pst) DC3000 infections [32]. Several calcium membrane transporters (calcium uniporter protein 2, mitochondrial, XM_003611866.2; putative calcium-transporting ATPase 13, plasma membrane-type, XM_013595727.1) and a transcription activator (calmodulin-binding transcription activator, XM_013591346.1) are upregulated, likely as a consequence of plant defense reactions (Figure 4b and Appendix A, *M. truncatula*). As reported above, maintaining calcium homeostasis is necessary to contain an excessive defense response to the infection, as the intracellular transport of this cation together with the production of reactive species are among the key events occurring during cellular immune signaling. Such a requirement of protecting tissues from an uncontrolled response is demonstrated by the strong downregulation of phosphatases (inorganic pyrophosphatase 2, XM_003608721.2, XM_013588831.1, XM_013588833.1, XM_013597279.1, XM_013588832.1, and XM_013586911.1), as similarly found in the mapping to *M. sativa* transcriptome.

### 2.4. GO Analysis of M. truncatula by SEA and PAGE Analysis

Four hundred sixty-four out of 671 upregulated transcripts with log2 fold change > +1 were annotated, of which 57 significantly enriched GO terms were identified with AGRIGoV2 using the *M. truncatula* V4.0 (JCVI) as reference genome. Biological processes with enriched GO terms identified by SEA analysis (Figure 5a and Appendix A, AGRIGoV2 results) were those related to Signal transduction (GO:0007165, 30 transcripts), Calcium ion transmembrane transport (GO:0070588, 6 transcripts) and metabolic processes, particularly related to protein amino acid phosphorylation and modification (GO:0006468, GO:00436687, 115 transcripts) (Figure 5b and Appendix A, AGRIGoV2 results).

One hundred forty-nine out of 264 downregulated transcripts with a log2 fold change <−1 were annotated, of which 18 GO terms were found significantly enriched. Two main biological processes were affected by *Xylella* infection, namely oxidation reduction (GO:0055114, 28 transcripts) and carbohydrate metabolic processes (GO:0005975, 19 transcripts) (Figure 5b and Appendix A, AGRIGoV2 results). 

When a search for biological processes with GO-enriched terms was carried out by a PAGE analysis that included log2 fold change values >+1, the only significant GO terms were defense response (GO:0006952, 28 transcripts) and response to stress (GO:0006950, 30 transcripts). Indeed among the more represented upregulated transcripts in this enriched category, the above reported TMV resistance protein N and several putative disease resistance proteins were found (Appendix A, AGRIGoV2 results).

Conversely, downregulated transcripts (log2 fold change < −1) belong mainly to significant GO:0016791, phosphatase activity (13 transcripts). 

## 3. Discussion

The high level of phenotypic diversity displayed in susceptible hosts upon infections with *X. fastidiosa* is a typical trait of a generalist pathogen that has a wide genetic variability (six subspecies and more than 80 sequence types have been so far described) whose role in disease severity is not completely understood. 

Among the *X. fastidiosa*-induced diseases, those caused by isolates of the subspecies *pauca* and harboring the genotype ST53 are particularly severe. This genotype has been reported to cause deadly infections in susceptible olive cultivars [10] and in other hosts like oleander and *Acacia* spp. (D. Boscia, personal observations). Surprisingly, ST53 isolates are not capable of causing infections in citrus, notoriously susceptible to strains of the subspecies *pauca* [33], nor even in grapevine. In *Prunus* spp., infections have been reported only in cherry and almond trees, although with mild symptoms. 

When we inoculated *M. sativa*, very severe reactions were consistently observed in the inoculated stems, which progress with necrosis and complete desiccation, a plant response that tends to but is not able to restrict the systemic movement of the pathogen, as shown by the finding that only in some of the plants the bacterium was detected in the non-inoculated portions, as well as in the roots. Previous studies using the same host but bacterial strains recovered from almonds and grapes report the development of symptoms of stunting and dwarfing, the so-called alfalfa dwarf (AD) disease, with grape strains eliciting more severe dwarfing phenomena than those isolated from almonds [34]. The overactive immune response detected upon the infections of this host with the olive-infecting bacterial strain clearly indicates that the bacterial genome information plays an important role for the expression of symptoms. 

The results of mapping to the transcriptome database of the same species and the genome assembly of the related *M. truncatula* were unambiguous, as both analyses identified similar differentially expressed genes and pathways involved. Upon infections with *X. fastidiosa*, alfalfa tissues suffer a severe biotic stress demonstrated by the activation of major plant immunity pathways, such as those, among the other, showing the altered expressions of genes related to calcium metabolism. Calcium’s role in immune response is mainly connected to signaling, as its cytoplasmic concentration increases after pathogen recognition at the membrane level by pattern recognition receptors (PRRs). Upon pathogen-associated molecular pattern (PAMP) or damage-associated molecular pattern (DAMP) recognition by PRRs, the influx of Ca++ ions in the cytoplasm is initiated. This cytoplasmic Ca++ increase led to the activation of several calcium-dependent protein kinases that transfer the signal to the nucleus to start an immune response [24,25]. Several genes supporting this process, from Ca++ transmembrane transport to Ca++-dependent transcriptional activators, are all upregulated in the infected alfalfa plants. An increase in cytoplasmic Ca++ is suddenly followed by the extracellular production of reactive oxygen species (ROS) to contrast pathogen replication. In *Arabidopsis thaliana*, a main link between the two processes is represented by the phosphorylation of the membrane-located NADPH oxidase (RESPIRATORY BURST OXIDASE HOMOLOGUE PROTEIN D, RBOHD), responsible for the production of ROS after PRR stimulation by PAMPs or DAMPs [26]. In our pathosystems, we found confirmation of this linkage, as shown by the upregulation of different RBOHD homologues (contig_99663 and contig_8164 in *M. sativa*; XM_013591674.1 and XM_013591675.1 in *M. truncatula*; Appendix A, DESeq2 analysis). In addition, the well-known TMV resistance protein N is also upregulated. In the *N. tabacum* pathosystem, its main role is to constrain the pathogen growth by inducing the hypersensitive reaction, or a localized cell death to confine the pathogen in the infection site and limiting its further spread. Among the defense processes associated with the hypersensitive response, the production of ROS has been already described. Further players of the cell death phenomenon are genes encoding lipid acid hydrolases belonging to the patatin-like gene family (PLP), which are upregulated. The expression of these enzymes with lipolytic activity toward membrane-associated lipids is induced by several fungal and bacterial pathogens and facilitates their spread by degrading tissues on the front of the infection [21]. 

All these reactions went, however, uncontrolled, as shown by the necrotic reaction of the inoculated stems. Alfalfa plants try to control this overactive immune response in multiple ways, as demonstrated by the involvement of genes necessary to maintain the redox status homeostasis, to regulate the phosphorylation status of the infected plants and to contrast the cell-wall degradation activity of *Xylella*. Containment of ROS production and scavenging is carried out by the Monothiol glutaredoxin-S6 (GRX S6) and a protein of the Thioredoxin family, both upregulated in infected tissues, while the need to maintain a phosphate homeostasis is shown by the downregulation of several phosphatases, as phosphorylation is a major regulating/signaling mechanism of enzyme status during immune response. Indeed, a xyloglucan-specific endo-β-1,4-glucanase inhibitor protein (XEGIP) was found to be strongly upregulated, most likely related to the attempt of the plant to inhibit *Xylella*-produced endo-β-1,4-glucanases, responsible for the degradation of the cell wall pit membranes that facilitate the bacterial movement and its pathogenicity [35]. 

These studies show that *M. sativa* can be a suitable host for dissecting *Xylella*–plant interactions as it is, particularly for the “De Donno” strain from olives, very susceptible to infections. The lag time before symptoms appear indicates that the pathogen needs to reach a threshold population abundance to be recognized by the host tissues before mounting an immune response. Moreover, the specific necrotic symptoms elicited by this *Xylella* strain with respect to those observed in AD disease, would allow for the identification of potential bacterial virulence factors and potentially for the translation of this information to olives, whose pathosystems are much more complex.

## 4. Materials and Methods 

### 4.1. Test Plants and Bacteria Inoculations 

Four-month-old seedlings of alfalfa were needle-inoculated with a highly concentrated bacterial suspension (>10^9^ CFU/mL) prepared by scraping bacterial cells from 7-day-old BCYE-grown cultures of the reference olive strain “De Donno.” Briefly, droplets of 5–8 microliters of a freshly prepared PBS–bacterial suspension were placed on the basal portions of the shoots, and punctured 3–4 times with an entomological pin to allow the bacterium to enter the stem. Two inoculation points for each stem were made [12]. Experiments consisted of two independent sets of plants inoculated in March 2018 (13 plants) and August 2018 (14 plants). In each experiment, three additional plants were inoculated with PBS 1X (KH_2_PO_4_ 1 mM, Na_2_HPO_4_ 3 mM, NaCl 155 mM) and used as mock-inoculated controls. 

Plants maintained at 25–28 °C were periodically inspected for symptoms, and tissues from different portions of the plants (leaves from the inoculated shoots and leaves from the non-inoculated shoots and roots) were collected at 3 and 6 mpi. 

### 4.2. Diagnostic Tests for Xylella fastidiosa

At least 0.2 g of tissues were used to recover the total DNA from each sample. Plant materials were macerated in Bioreba extraction bags with Lysis buffer (1:10 *w*:*v*), and DNA purified using the DNeasy Food Mericon kit (Qiagen, Valencia, CA, USA). An aliquot of 2 µL of the purified DNA was added to the real-time PCR reactions using the primers and probe designed by Harper et al. (2010) [36]. 

To further verify the presence of actively growing bacterium in the plants that showed systemic colonization (i.e., positive qPCR results obtained for the samples collected from the non-inoculated shoots and from the roots), isolation in pure culture was also attempted from the leaves of these plants. For the isolation, leaves were surface sterilized with sodium chloride (2%) and ethanol (70%), prior to be macerated in Bioreba extraction bags using PBS 1X (1:5 *w*:*v*). Aliquots of 100 µL of the recovered 10-fold serial dilutions corresponding to 1:100 to 1:1000 were then plated in a periwinkle wilt gelrite (PWG) medium. Plates were incubated in the dark at 28 °C for 2 months. 

### 4.3. Purification of Total RNAs for High Throughput Sequencing

Three of the four plants yielding positive qPCR reactions in both roots and non-inoculated roots were selected for the transcriptome analysis (EM_1, EM2, and EM3), while three mock-inoculated seedlings (EM_H1, EM_H2, and EM_H3) were selected for comparison. Total RNAs were extracted from stems and leaves combining a guanidium thiocyanate-phenol-chloroform-extraction with purification on the RNeasy Plant Mini kit (Qiagen, Valencia, CA, USA). One gram of plant tissue was pulverized in liquid nitrogen prior to adding (i) 5 mL of extraction buffer (4 M guanidine thiocyanate, 0.2 M sodium acetate, pH 5.0, 25 mM EDTA, 2.5% (*w*/*v*), and polyvinylpyrrolidone (PVP) and (ii) 500 μL of 20% N-Lauroylsarcosine sodium (NLS). Five milliliters of the sap were aliquoted into 2 mL collection tubes (1 mL each) and incubated at 65 °C temperature for 10 min and then cooled in ice. Samples were sequentially extracted with one volume of phenol-chloroform and then chloroform by vortexing and centrifuging at 16,000× *g* for 10 min at 4 °C. Total RNAs were then precipitated from the upper phase by adding 0.1 vol NaOAc (3 M; pH 5.2) and 2.5 vol of 100% EtOH and centrifuging at 14,000 rpm at 4 °C for 20 min. Water-resuspended total RNAs were further purified by RNeasy Plant Mini Kit (QIAGEN, Valencia, CA, USA) and DNAse digested following the manufacturer’s instructions. Poly A enrichment of total RNAs, Illumina TruSeq RNA library construction, followed by 2 × 100 NovaSeq sequencing were outsourced to Macrogen Inc. (Seoul, Korea). 

### 4.4. Analysis of Differential Gene Expression, Pathway Involvement, and Gene Ontology

The TopHat2 program [37] was used to calculate gene expression levels in each library by quantifying the number of Illumina reads that mapped to the de novo transcriptome assembly (MSGI 1.2; [15]) of the subspecies *sativa* (B47) and *falcata* (F56) and to the *Medicago truncatula* genome assembly GCF_000219495.3_MedtrA17_4.0 [15]. Raw counts of the mapped reads were extracted from the TopHat2 alignments from each library to estimate the gene/transcript abundance using the SeqMonk v1.45.0 program (https://github.com/s-andrews/SeqMonk). The DESeq2 Rpackage version 1.24.0 [38] was used to statistically evaluate the relationships among different libraries and the differential expression level of the genes/transcripts. DESeq2 performs statistical analysis (calculation of sample-to-sample distances; PCA) on raw counts to execute normalization, variance estimation, and differential expression. The software uses a model based on negative binomial distribution by adjusting the obtained P-values by Benjamini and Hochberg’s procedure for controlling the false discovery rate. Stringent parameters consisting of a false discovery rate (FDR) of p-values less than 0.001 were used in DESeq2 to identify differential expressed genes. DEG annotation was performed by downloading the “contig annotation” database from The Alfalfa Gene Index and Expression Atlas Database (http://plantgrn.noble.org/AGED/Download.jsp).

Transcripts categorization was also performed by MapMan v.3.5.1 [39] using FASTA files containing all differentially expressed genes. Statistically enriched bins/pathways (*p* < 0.05) were identified by a Wilcoxon rank sum test followed by Benjamini–Hochberg correction for multiple testing.

Gene ontology analysis was performed by the AgriGov2 gene analysis tool, which supports the Medicago truncatula genome assembly (http://systemsbiology.cau.edu.cn/agriGOv2/) [40]. Singular enrichment analysis (SEA) tool identifies GO-enriched terms that are statistically significant (*p* < 0.05) with respect to a pre-calculated background of terms. GO enrichment studies were also performed by Parametric Analysis of Gene Sete Enrichment (PAGE) that includes the gene expression levels expression (log 2 fold change) in the analysis.

### 4.5. Data Records

The raw data have been deposited into NCBI’s Sequence Read Archive (SRA) under the BioProject PRJNA563361, BioSample SAMN12672938, and SRA IDs SRR10056634, SRR10056633, SRR10056632, SRR10056631, SRR10056630, and SRR10056629.

## Figures and Tables

**Figure 1 plants-08-00335-f001:**
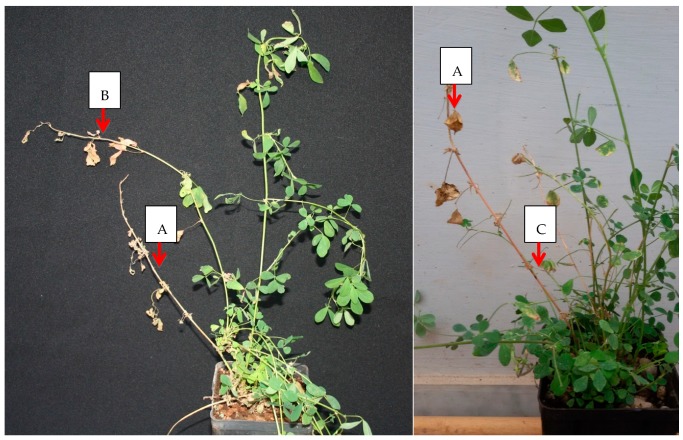
Plants of *Medicago sativa* inoculated with *Xylella fastidiosa* strain “De Donno” at four months post inoculation. Arrows indicate the desiccation phenomena recorded on two inoculated shoots: (**A**) shoot completely desiccated; (**B**) shoot showing the initial stage of the desiccation phenomena (apex dieback); (**C**) inoculation point.

**Figure 2 plants-08-00335-f002:**
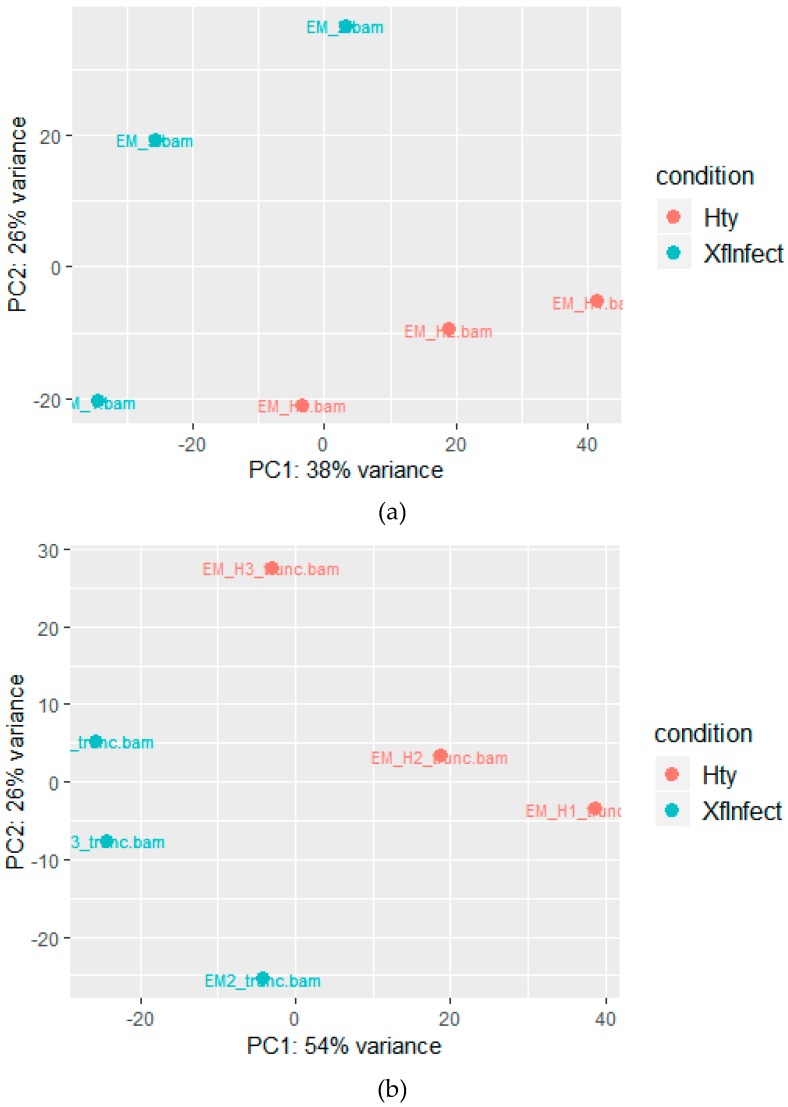
Principal component analysis (PCA) of reads mapped to (**a**) the MSGI 1.2 transcriptome database or (**b**) the *M. truncatula* genome.

**Figure 3 plants-08-00335-f003:**
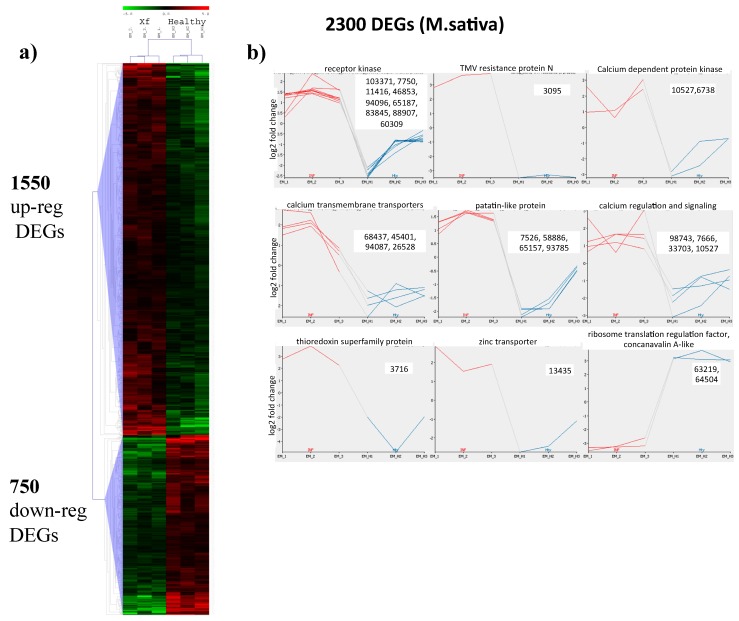
Differentially expressed genes in *Medicago sativa* infected by *Xylella fastidiosa*. Heat map representation (**a**) of up (red) and down (green) DEGs. (**b**) Graphs showing the individual log2 fold change expression of selected genes from *Xylella*-infected (red) or healthy (blue) plants. Numbers refer to the contig’s identifiers as reported in the text.

**Figure 4 plants-08-00335-f004:**
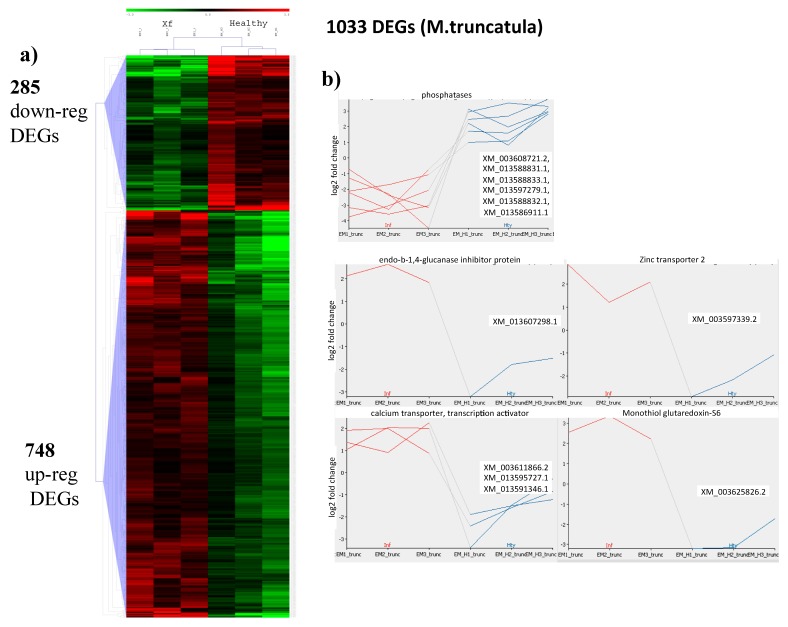
Differentially expressed genes in *Medicago truncatula* infected by *Xylella fastidiosa*. Heat map representation (**a**) of up (red) and down (green) DEGs. (**b**) Graphs showing the individual log2 fold change expression of selected genes from *Xylella*-infected(red) or healthy (blue) plants upon *Xylella* infection. Identifiers (XM_#) correspond to the mRNAs annotated in the MedtrA17_4.0 version of *M. truncatula* genome as reported in the text.

**Figure 5 plants-08-00335-f005:**
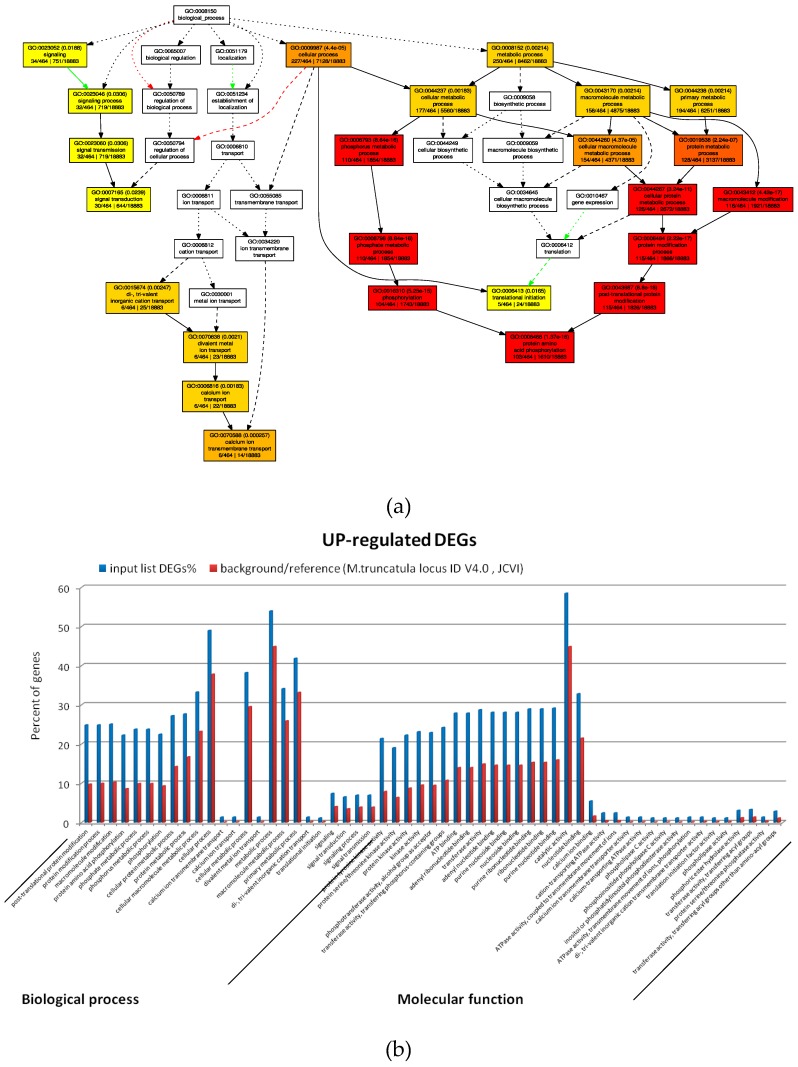
GO analysis by SEA. Graphical representations of biological and metabolic processes showing enriched GO terms among the *Medicago truncatula* upregulated (**a**) and downregulated (**c**) transcripts obtained by SEA analysis. Statistical significance (*p* < 0.05) is indicated by p-values of each term (*p* < 0.05) and the color (i.e., from yellow to red the significance increase). Graphs showing the 57 upregulated (**b**) and 18 downregulated (**d**) GO terms detailed by biological process and molecular function. Enriched DEGs (blue bars) are compared to the background expression (red bars) of the *M. truncatula* V4.0 (JCVI) as a reference genome.

**Table 1 plants-08-00335-t001:** Results of the needle-inoculations of *Xylella fastidiosa* strain “De Donno” on *Medicago sativa.*

First Experiment—Inoculation Date: 30/03/2018						
	3 Months Post Inoculation				6 Months Post Inoculation *	
**N. inoculated plants**	Inoculated shoot n. positive plants (average Cq values)	Non inoculated shoot n. positive plants (average Cq values)	Symptoms	Non inoculated shoot n. positive plants	Roots: n positive plants	Symptoms
13	6 (29,27)	0/13	Leaf scorching	0	0	3 plants showed desiccated inoculation shoots (Figure 1)
**Second Experiment—Inoculation Date: 21/08/2018**						
14	7 (21.26)	0	Initial shoot apex dieback on 6 plants	7 (26,66)	4 **	8 plants showed desiccation of the inoculation

* At 6 mpi, the assays were also repeated for the inoculated shoots testing negative at 3 mpi; qPCR reactions yielded consistent negative results. ** The four plants yielding positive qPCR reactions in the roots were comprised among the 7 plants yielding positive reactions on the foliage collected from the non-inoculated shoots.

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
