# Peer review of "Infections of the Xylella fastidiosa subsp. pauca Strain “De Donno” in Alfalfa (Medicago sativa) Elicits an Overactive Immune Response"

_plants, 2019, doi:10.3390/plants8090335_

Round 1
Reviewer 1 Report
Kubaa et al. reported an Infections of Xylella fastidosa subsp pauca strain “De Donno” in Alfalfa (Medicago sativa) elicits an overactive immune response. In the present manuscript, authors have inoculated alfalfa (Medicago sativa) with the olive-infecting strain “De Donno” isolated from a symptomatic olive in Apulia (Italy), and demonstrated that this highly pathogenic strain causes an overactive reaction that ends up with the necrosis of the inoculated stem, a reaction that differs from the notoriously Alfalfa Dwarf disease, caused by X.fastidiosa strains isolated from grapes and almond. This study holds a great promise from the overall objective in the biotic stress factor and from protective plants from these diseases. Though the structure of the study is well planned but the results are not so descriptive and not very well illustrated. Results need significant improvement and the authors need to put the best results in the manuscript. The discussion needs to be more specific and also specify and discuss the genes identified, rather than giving a very vague and overall discussion in this study. To make the findings more interesting to the scientific community major changes are required in the manuscript
Major questions and comments that should be addressed.
Results: Figure 1. A clearer picture would have been better. Moreover, desiccated shoot, inoculation point and shoot structure could have been illustrated more significantly by magnifying those areas or cutting those parts and clicking pictures in petri plates. In the title of the manuscript “fastidosa” spelling is wrong. It should be fastidiosa Results: Page 4 line 127 to 146 – Authors have shown that they identified 2300 DEGs and 1500 upregulated and 750 down-regulated, it would be great to show some Venn diagrams, heat maps, or bicluster plots for this DEG up and down ones. The representation of the data is also very important. Only supplementary files do not provide everything, results should be made more promising by providing good figures. Results: Page 4 line 154 to 162: Again authors need to describe these results as well. 619 DGs need to be described and represented in Venn diagrams, heat maps or bicluster plots. Even bar graphs for functional categorization will also help to represent this data. Results: Page 5 line 182 to 185: Reframe this sentence. Make this sentence clear with proper grammar and technical aspect. Results: Page 5 line 182 to 185: Reframe this sentence. Needs clarification. Table 1 page 5: I think this table is not required. Authors have already mentioned everything in the materials and methods section. Figure 2 page 6: Legends needs to be enlarged and the resolution of the figure needs to be improved. Figure 3 page 8: enlarge the figure 3 a, describe the colour code used properly in the legend of the figures.Minor revisions:
All scientific names in italics Figure 2. Legend, line 349. Check it throughout the manuscript. Put references in the discussion sections. Put references in the materials and methods lines 338-345, section 4.2 lines 350-360. GO enrichment can be provided in the form of a bar graph as well with functional analysis. Please mention the PBS concentration used in the inoculation media in the materials and method section. Spell check the full manuscript Line 375 its degree Celsius (°C) Line 168: and Check for grammatical errors, spell check and punctuations throughout the manuscript before final submission.
Author Response
Kubaa et al. reported an Infections of Xylella fastidosasubsp pauca strain “De Donno” in Alfalfa (Medicago sativa) elicits an overactive immune response. In the present manuscript, authors have inoculated alfalfa (Medicago sativa) with the olive-infecting strain “De Donno” isolated from a symptomatic olive in Apulia (Italy), and demonstrated that this highly pathogenic strain causes an overactive reaction that ends up with the necrosis of the inoculated stem, a reaction that differs from the notoriously Alfalfa Dwarf disease, caused by X.fastidiosastrains isolated from grapes and almond. This study holds a great promise from the overall objective in the biotic stress factor and from protective plants from these diseases. Though the structure of the study is well planned but the results are not so descriptive and not very well illustrated. Results need significant improvement and the authors need to put the best results in the manuscript. The discussion needs to be more specific and also specify and discuss the genes identified, rather than giving a very vague and overall discussion in this study. To make the findings more interesting to the scientific community major changes are required in the manuscript.
We thank the reviewer for these remarks, we have now added several illustrations and more details. We hope we addressed her/his main suggestion and completely satisfy her/his requests.
Major questions and comments that should be addressed.
Results: Figure 1. A clearer picture would have been better. Moreover, desiccated shoot, inoculation point and shoot structure could have been illustrated more significantly by magnifying those areas or cutting those parts and clicking pictures in petri plates.Response: we have more pictures, however we have already selected those that can show the whole and general status of the inoculated plants with both desiccated and non-symptomatic shoots.
In the title of the manuscript “fastidosa” spelling is wrong. It should be fastidiosaResponse: the title was corrected
Results: Page 4 line 127 to 146 – Authors have shown that they identified 2300 DEGs and 1500 upregulated and 750 down-regulated, it would be great to show some Venn diagrams, heat maps, or bicluster plots for this DEG up and down ones. The representation of the data is also very important. Only supplementary files do not provide everything, results should be made more promising by providing good figures.Response: We believe that Venn diagram is not very appropriate for a clear representation of the Infected/Healthy conditions. Rather, we choose to present the data by a heat map and decided to describe the fold expression of selected genes as discussed in the text. We think that this option gives the reader a clearer representation of the whole transcriptome analysis and of the major biochemical pathways involved in the immune response (Figure 3). We present in this format also the transcriptome data of the M. truncatulamapping (Figure 4).
Results: Page 4 line 154 to 162: Again authors need to describe these results as well. 619 DGs need to be described and represented in Venn diagrams, heat maps or bicluster plots. Even bar graphs for functional categorization will also help to represent this data.Response: MapMan software has not the function of presenting Venn, diagram, heat maps or bicluster plot. The software can provide general or specific (secondary metabolism, hormone, signaling etc) pathways in a graphical format. We think that, as the MapMan analysis largely confirm the data from the DeSeq and GO analysis, we can omit to add additional graphical representations that, according to the MapMan significant categories are 6: “Pathways containing signaling (43), development (18), RNA (18), transport (23) and protein metabolism (17) transcripts, all having statistical significance at p<0.05”, as already reported in the text and Supplementary Table 3.
Results: Page 5 line 182 to 185: Reframe this sentence. Make this sentence clear with proper grammar and technical aspect.Response: we changed and improved the sentence according to the suggestion.
Results: Page 5 line 182 to 185: Reframe this sentence. Needs clarification.Response: we changed and improved the sentence according to the suggestion.
Table 1 page 5: I think this table is not required. Authors have already mentioned everything in the materials and methods section.Response: we prefer to maintain the Table as it visually describe the output of the inoculations trials.
Figure 2 page 6: Legends needs to be enlarged and the resolution of the figure needs to be improved.Response: we enlarged the legends and figure resolution
Figure 3 page 8: enlarge the figure 3 a, describe the colour code used properly in the legend of the figures.Response: we enlarged the legends and figure resolution and changed the description of the colour code of the figure
Minor revisions:
All scientific names in italics Figure 2. Legend, line 349. Check it throughout the manuscript.Response: Names were changed in italic, also in legend.
Put references in the discussion sections.Response: we improved the discussion and included references
Put references in the materials and methods lines 338-345, section 4.2 lines 350-360.Response: we included the reference
GO enrichment can be provided in the form of a bar graph as well with functional analysis.Response: GO enrichment by bar graph was included in Figure 5.
Please mention the PBS concentration used in the inoculation media in the materials and method section.Response: PBS recipe was included
Spell check the full manuscriptResponse: we checked the whole manuscript
Line 375 its degree Celsius (°C)Response: we did the correction
Check for grammatical errors, spell check and punctuations throughout the manuscript before final submission.Response: we have carefully re-checked the manuscript

Reviewer 2 Report
Congratulations for your report. It is not easy to set up new plant-pathogen systems in the lab as a model of study.
However, to my opinion, there are 2 elements of information missing:
In lines 72-74 you say that you did several attempts to infect several species. I think would be useful for other researchers to know in which species it didn't work, as supplementary information or in just one tense included in"results". You make reference to your last paper (12,Giampetruzzi et al., 2017) but that paper is just a short announcement of the genome of the strain of xilella fastidiosa you used. There is no information about the fastq files generated being available in any public database so other researchers can double-check your results or use them for other research pourpouses.Minor revision of English:
-line 108: information is (not are) lacking. Actually I would rephrase the whole tense.
-line 166: "orthologues" not "hortologous"
Author Response
Reviewer 2
Authors
Congratulations for your report. It is not easy to set up new plant-pathogen systems in the lab as a model of study.
However, to my opinion, there are 2 elements of information missing:
In lines 72-74 you say that you did several attempts to infect several species. I think would be useful for other researchers to know in which species it didn't work, as supplementary information or in just one tense included in"results".
Response: the name of the species we tried to infect are now included.
You make reference to your last paper (12,Giampetruzzi et al., 2017) but that paper is just a short announcement of the genome of the strain of xilella fastidiosa you used. There is no information about the fastq files generated being available in any public database so other researchers can double-check your results or use them for other research purposes.
Response: The reference Giampetruzzi et al 2017 is included to provide information on the strain used in our experiment: the reference strain “De Donno” of X. fastidiosa, subspecies pauca ST53. Genome Announcements did not require to release the fastq files. It is not clear to us this comment, since the genome announcement refers to the bacterial genome, while this work relates to the RNAseq of the infected plants and the assessment of the fastq files would not affect the data included in this manuscript.
Minor revision of English:
-line 108: information is (not are) lacking. Actually I would rephrase the whole tense.
Response: we have edited the sentence.
-line 166: "orthologues" not "hortologous"
Response: we did the correction

Round 2
Reviewer 1 Report
I am satisfied with the changes provided.
Author Response
Dear Reviewer,
we deposited the RNAseq data in the NCBI database. We reported the IDs of data deposit in a new chapter, 4.5 Data Records.
thanks for your help and advices.
kind regards
Pasquale Saldarelli
